# Measurement System for Unsupervised Standardized Assessments of Timed Up and Go Test and 5 Times Chair Rise Test in Community Settings—A Usability Study

**DOI:** 10.3390/s22030731

**Published:** 2022-01-19

**Authors:** Sebastian Fudickar, Alexander Pauls, Sandra Lau, Sandra Hellmers, Konstantin Gebel, Rebecca Diekmann, Jürgen M. Bauer, Andreas Hein, Frauke Koppelin

**Affiliations:** 1Assistance Systems and Medical Device Technology, Carl von Ossietzky University Oldenburg, 26129 Oldenburg, Germany; sandra.hellmers@uni-oldenburg.de (S.H.); konstantin.gebel@gmail.com (K.G.); rebecca.diekmann@uni-oldenburg.de (R.D.); andreas.hein@uni-oldenburg.de (A.H.); 2Institute of Medical Informatics, University of Luebeck, 23538 Luebeck, Germany; 3Section Technology and Health for Humans, Jade University, 26121 Oldenburg, Germany; frauke.koppelin@jade-hs.de; 4Center for Geriatric Medicine, University Heidelberg, 69117 Heidelberg, Germany; sandra.lau@uni-heidelberg.de (S.L.); Juergen.Bauer@bethanien-heidelberg.de (J.M.B.)

**Keywords:** Timed Up and Go Test, 5 Times Chair Rise Test, sit to stand test, assessment, unsupervised, functional, system usability, SUS, evaluation

## Abstract

Comprehensive measurements are needed in older populations to detect physical changes, initiate prompt interventions, and prevent functional decline. While established instruments such as the Timed Up and Go (TUG) and 5 Times Chair Rise Test (5CRT) require trained clinicians to assess corresponding functional parameters, the unsupervised screening system (USS), developed in a two-stage participatory design process, has since been introduced to community-dwelling older adults. In a previous article, we investigated the USS’s measurement of the TUG and 5CRT in comparison to conventional stop-watch methods and found a high sensitivity with significant correlations and coefficients ranging from 0.73 to 0.89. This article reports insights into the design process and evaluates the usability of the USS interface. Our analysis showed high acceptance with qualitative and quantitative methods. From participant discussions, suggestions for improvement and functions for further development could be derived and discussed. The evaluated prototype offers a high potential for early detection of functional limitations in elderly people and should be tested with other target groups in other locations.

## 1. Introduction

Muscle weakness, reduced gait speed, and fear of falling are strong predictors for developing functional disabilities causing inactivity [1,2] and restrictions in activities of daily living (ADL) [3]. Several limitations in the ability to handle ADLs may also lead to a higher mortality rate [4]. Maintaining physical activity and exercise in older populations can reduce or reverse lost muscle mass [5], increase physical capacity and quality of life [6], and preserve cognitive and intellectual status [7,8]. Physical activity can further reduce conditions associated with frailty [9,10]. Therefore, comprehensive and continuous measurements are necessary to detect functional changes and initiate early interventions to avoid physical deterioration and loss of mobility [11,12].

Existing instruments such as the Timed Up and Go (TUG) [13] and the 5 Times Chair Rise Test (5CRT) [14] are commonly used to assess corresponding functional parameters in geriatric care. Both have sensitive predictors for disability [15] and recurrent falls [16]. The TUG measures the time it takes for an elderly individual to stand up from a chair, walk three meters, turn around, walk back, and sit down. It demonstrates moderate to good sensitivity for predicting fall risks [17], Parkinson’s disease [18,19], and balance disorders [20]. As part of the short physical performance battery (SPPB) [21], the 5CRT is well adapted for assessing leg power [22]. To perform the 5CRT, the participant sits centered on a chair, placing his arms across the chest (for a detailed description see [14]). Stopwatches are typically used to perform time measurements in clinical settings.

To minimize measurement errors and increase retest reliability, automated procedures were introduced using technical screening systems like the ambient TUG (aTUG) chair and IMU-based wearable sensors. Both show a high correlation with conventional assessments of TUG [23,24] and 5CRT [14,25] and are thus appropriate for the early detection of functional changes.

Besides test sensitivity, the early detection of functional decline requires frequent screening, which can be challenging to achieve in healthcare settings. While regular and brief screenings by physicians or physical therapists (e.g., yearly screenings for fall risks) are suggested by the American Geriatrics Society [26] and offer a pragmatic approach for performing time-framed examinations, individual assessment frequencies might be necessary to detect and observe early changes of physical function. Thus, regular assessments are essential. Given the time constraints faced by therapeutic and health professionals, and the resulting demand for documentation and administration, it seems unlikely that routine contact provides the necessary setting to detect these changes.

Sensor-based measurements initiated by the older adults provide a more suitable path with greater potential. In theory, the optimal screening method would be a continuous data collection obtained, for example, through wearable devices. While various systems have been proposed for the extraction of stair-climb power [27], and mobility in general [28,29], the monitored biomechanical parameters are most meaningful within a stable context. Thus, a monitoring system for home use might be susceptible to unrecognized contextual variations [28]. A consequent assessment in a standardized setting (e.g., within a clinical screening) to assure high intertest reliability is more suitable.

While most of the corresponding technical screening tools are well suited for guided assessments (e.g., in community settings), they still require the (tele-)presence of physicians or therapists, as their independent use by older populations may present a challenge. The correct execution of movements can also be affected, despite being mandatory for the comparability of results.

As frequent monitoring (e.g., monthly) of early functional changes requires high levels of organizational and financial effort, individualized care by experts seems inapplicable. The unsupervised screening system (USS) [30] overcomes the need for trained experts, provides a sensitive sensor to measure TUG and 5CRT, and enables regular unsupervised testing for older individuals in community settings and therapeutic rehabilitation.

The use of automated assessments by technical screening systems such as the ambient TUG (aTUG) has been shown to be sensitive to the detection of functional impairment and should also increase reliability between successive measurements [23]. Although the aTUG still requires the presence of a supervisor, this already has automatic measurement capabilities. Via an infrared light barrier (LB), four force sensors (FS), and a laser range finder (LRS), the aTUG measures the TUG. As proposed by Botolfsen et al. (2008), the aTUG can automatically measure the total time of the TUG and all subtasks of the instrumented TUG (iTUG) using these sensors [31].

The recently proposed short physical performance battery (SPPB) kiosk [32] is designed for supervised evaluation of the SPPB with its three components (gait speed, 5SST, and standing balance) and is intended to improve intertest reliability in performing the SPPB protocol. Semiautomatic postprocessing was used to demonstrate the validity of the SPPB kiosk for estimating these SPPB components. The applicability of inertial measurement units (IMU) to automatically measure TUG and 5SST performance by measuring acceleration and gyroscopic rotation rate was similarly confirmed [14,25].

To assure willingness of frequent use, these systems must address aspects of usability and acceptance alongside the sensitivity of functional measures. Among the existing factors for technology acceptance in older populations, the perception of usefulness and potential benefit (as value), user-friendliness and ease of learning (as usability), and feeling of empowerment without anxiety or intimidation (confidence) are relevant [33].

Usability relates to perceptions of user-friendliness and ease of learning. Older adults who are aware of a system’s technological benefits and are willing to try new procedures [34] are more likely to adopt and continue to use tools that help them remain independent [33]. However, as the consideration of an older adult’s ability differs from the general population, both physically and cognitively, and familiarity with new technology, including technology literacy, computer anxiety is important for appropriate system design.

It is thus essential to focus on technological benefits. The perceived ease of understanding, utility, and use are key determinants of adoption [35,36]. It must also be noted that, when faced with unfamiliar technology [37], older adults tend to express a lower level of familiarity and trust compared to younger groups, and dislike technology that requires too much effort to learn or use [38]. Users should consequently avoid confusion with excessive features, options, or information [38]. Interfaces should be intuitively understandable and manageable [39]. The use of touch screens, for example, may reduce workload by clearly matching display and control [40]. Consequently, the corresponding applicability of unsupervised, functional status screening systems by older adults has yet to be confirmed from a usability and acceptance standpoint.

This article addresses these aspects and holds the following contributions: (1) The user interface for a screening system for unsupervised assessment by older adults of the TUG and the 5CRT; (2) The screening system’s usability, user acceptance of the system, and the experience with it is studied in a two-stage development process.

## 2. Materials and Methods

To evaluate and enhance the usability of the USS, a two-step iterative participatory design process was applied. The findings of both evaluation cycles are presented beside the applied methodology. The initial prototype and corresponding usability study, described in Section 2.1, focused on the user interaction and motivational aspects of the USS, and addressed aspects of usability and acceptance from a functional and content-driven perspective. Based on the insights obtained (see Section 2.1.3), the user interface was enhanced, and measurement technology integrated, as described in Section 2.2.1. The usability of the resulting USS is evaluated in Section 2.2.

### 2.1. Initial Prototype and Usability Study

To guide users autonomously conducting the functional TUG and 5CRT assessments, an operational sequence of audiovisual instruction and user interactions were designed based on insights from the AEQUIPA Versa study [41,42]. This initial prototype aimed to investigate potential enhancements in the detail and framing of the assessment instructions and to assess whether they were appropriate for increased user coherence.

#### 2.1.1. USS’s Initial User Interface

To assure effective usability for the intended target group (65 years and above), the initial USS prototype (shown in Figure 1) was developed per age-related requirements, including perception and cognitive specifics [43]. It integrates all user interfaces and consists of an aTUG sensor chair [23], a sensor belt, shown in Figure 2, and a display. For user interaction (UI), a radio-frequency identification (RFID) reader, keyboard, and mouse were used. A full-HD display with a diagonal measurement of 106 cm and the UI devices were placed on a table that stands approximately 4 m in front of the aTUG chair (as shown in Figure 1).

The prototype implements designed **user interaction phases** and their corresponding workflows. As shown in Figure 3, potential users can inform themselves via an **introduction video** that motivates the USS’s use, introduces the assessments, and instructs the initial authentication process.

RFID tags are used for **user authentication** due to their confirmed usability for older adults [44,45] and compatibility with keyrings.

Following the initial authentication, users are asked to enter **registration** information (age, weight) via keyboard and mouse.

To benefit from older adults’ experience with TVs, media presenting video-based instructions for upcoming tasks is used. Subtitles are also included for those with hearing impairments, representing spoken instructions. With the **preparation phase**, for example, it is accompanied by a corresponding video tutorial, which indicates how to disconnect and fit the USS sensor belt. Even though we expect a later version of the system to rely on a wireless connection and charged sensor belt, the current prototype relies on a USB connection for charging and data transfer.

The TUG, 5CRT, and weighting **assessments** must then be conducted. For the conclusive prototype, the Mini Nutritional Assessment Short Form, a questionnaire that screens nutritional status and identifies malnutrition [46], was also incorporated.

After the assessment phase, users can **review their results** on dedicated charts. These are presented in a timeline against previous screening results (as shown in Figure 4). Within these charts, performance is color-coded via red-, yellow-, and green-based categories, shown on a common scale. For example, the overall TUG test duration, and that of its subphases, are shown under the instrumented TUG (iTUG) [47] guidelines. Users are expected to **complete the assessment** by logging out of the USS and rescanning their RFID chip.

#### 2.1.2. Initial Usability-Study

To investigate the usability of the first prototype and clarify user preferences, the initial study was structured into three parts. The first required participants to follow a general introduction of the study (explaining briefly the general purpose of the system and the common interaction phases via storyboards) without discussing the UI mechanisms. During a **task-based evaluation**, participants were then asked to follow prompts with no support and report their challenges and their experience by applying Nielsen’s thinking-aloud methodology [48]:Task 1 asked participants to “create a new user account and then log out of the system”. This covered implicitly, user comprehensibility, motivational aspects of the introductory video, and the suitability of the RFID login mechanism for authentication. The possibility that inputting age and weight parameters during the registration process may represent a barrier was also evaluated.Task 2 asked participants to “perform all mobility tests using the system and view your results at the end”. The applicability of the sensor belt was thus investigated, as well as potential challenges in user confidence resulting from unplugging and replugging in the USB cable. The clarity of the video instructions, regarding both auditory and visual perceptions, and the clarity of the explanation were evaluated.Task 3 required participants to “change your username and volume in the settings menu” and evaluated the usability of the corresponding functionality for parameter editing.

Finally, a **semi-structured interview** was conducted to investigate participants’ experiences with the USS. Herein, a questionnaire that addressed interface preferences and experiences was used. To quantify the perceived general usability of the system and motivational aspects of its use, participants had to fill in a 5-item Likert scale based on the system usability scale (SUS) [49]. Data was processed following Brooke (1996) [49]. By interpreting these results via the scale of Bangor et al. [50], it is possible to derive a rough trend on the usability of the overall system.

To clarify challenging components and user preferences, an additional questionnaire was conducted. This combined quantitative (5-item Likert scale) and qualitative questionnaire required textual feedback on specific aspects of user interaction, including video material, noise signal, and motivation for each UI phase (as shown in Figure 3). Individual preferences regarding these items were investigated (participants were requested to comment for self-explainability and suitability regarding participants’ ability to conduct the aforementioned tasks via the video instructions):The availability of subtexts.The preferred gender of the speaker.The suitability of the introduction video.The suitability of the multiple instruction videos.The suitability of the preparation videos.Preferences of the signaling tone indicating the start of an assessment.Assessment of the general suitability of the measurement system.Suitability of the presentation of the results.

The interview closed with general, open questions regarding perceived aspects of user interaction during using the USS. Each interview was audiotaped for later qualitative evaluation.

The study was approved by the General Ethics Committee of CvO University Oldenburg No. Drs 55/2017 and was conducted in accordance with the Declaration of Helsinki.

#### 2.1.3. Results and Discussion of the Initial Usability Study

For the initial usability study, 10 participants aged from 65 to 83 years (median 74.5, IQR 9) with a balanced gender distribution were interviewed in three iterative meetings. Despite the limited size of the study sample, the chosen evaluation approach is expected to provide sufficient feedback on interaction concepts requiring improvement.

Figure 5 shows a box plot with scores determined for all participants. The median SUS score of 75 indicates the generally good usability of the overall system. It can also be deduced that 25% of the ratings describe excellent usability. The lower whisker, measured at 65, shows that even the lowest score remains within the acceptable range. Using the results of the SUS, overall USS usability is confirmed.

The following usability challenges were identified (ordered by decreasing frequency of occurrence) and corresponding adjustments were subsequently made.

Some participants suggested that the video instructions were too lengthy and confusing. It was also pointed out that they focused on adults “much older” than the interviewees. By speaking clearly with distinct pauses, most participants felt the video presenter and general system addressed another target group. After identifying evidence for the well-known tendency of elderly groups not wanting to be addressed as restricted [33], we implemented a personalized approach. This offered both a short introductory audio with instructions and an extended video with instructions, activated once handling errors are recognized by the system.The USS’s limited responsiveness was identified as a major challenge. Reported contests with instructional videos indicated that some users struggled to distinguish the videos from the menu structures. Attempts to enter data while the instructional videos were still running, and user interaction was still blocked, did not contribute to the perceived user confidence. This became especially obvious in a longer instructional video among which no user interaction was supported. The recognized desire for shorter instructional videos combined with the challenges of the resulting delayed interaction induced the following approach: By replacing the video stream with combined explanatory audio and static visualization (shown in Figure 6), the new instructional method enables continuous user interaction (e.g., skipping the audio voice, or adding additional data) and, thus, should overcome this challenge. These video tutorials are also shown when a user fails to complete a test.The sensor belt was either not detached from the USB cable or closing the belt buckle was perceived as nonintuitive, presenting a partial challenge. It has since been adjusted to ensure usability and proper attachment among participants. As shown in Figure 2, the buckle mechanism has been replaced by Velcro to support the belts’ easy attachment and detachment. To “remind” users to disconnect the sensor belt from the USB connector, the USB cable was shortened, so disconnection became an automatism. The correct use of the sensor belt was also addressed in the preparation video, and correct and faulty examples have been added. In addition, the upper side is marked by a yellow string so the user can assure the belt’s correct attachment. For the convenience of participants with an increased waist width, the belt’s length was extended.The presentation of the introductory video inside of the system was deemed a mental barrier. Participants suggested that the initial motivation and authentication process take place outside of the system. To overcome the necessity of sitting down in the system, which may serve as a psychological barrier for initial users, an external display (running a motivational video) has been added and the authentication mechanism (RFID reader) moved outside.For the TUG, some participants neither started from the back of the chair nor ended with their backs leaning against it. Instead, users moved forward during the countdown and remained there until the test was complete. This might have created erroneous results. They also reported uncertainty on if/when the assessment had finished. While the 5CRT was well performed in most cases, participants sporadically stopped with less than five repetitions, resulting in invalid assessments. As a solution, missed performances during the assessment period have been covered with hints in the instruction phase. These indicate potential errors through positive/negative examples. For the 5CRT, an audio signal has been added that notifies users once five sit-to-stand cycles have been completed.Some participants ignored the instructions during the logout process and missed the announcement regarding the intended frequency of the system use. To remedy this issue, the announcement has been moved in front of the log-out screen.

In addition to the aforementioned challenges, some factors require future consideration. The interpretation of the result visualization was challenging for most participants. While the color-encoding approach was sufficiently intuitive, the visualization screen was only partly self-explanatory. Optimization potential can thus be given, for example, by integrating additional simplified charting summaries and explanatory videos.

### 2.2. Conclusive Evaluation Prototype and Study Design

The conclusive prototype, described in Section 2.2.1, implements insights from the initial usability study, integrates measurement technology (a sensor belt and ambient sensors, see [30] for further details), and acts as a foundation of the conclusive usability study (Section 2.3).

#### 2.2.1. Conclusive Evaluation Prototype

With insights from the usability study and first USS prototype (see Section 2.1.3), the evaluation prototype (shown in Figure 7) has been enhanced. The general USS cabin was constructed from wood. For extra privacy, curtains were placed in the openings which can be closed by participants. The USS has a total size of 2.30 m in height, 1.50 m in width, and 5.50 m in length. The sensor technology (aTUG, sensor belt, and two RGB-D cameras) has also been integrated.

By placing the display next to the user with a touch-based system, participants can interact directly and overcome the use of a keyboard and mouse. The user is thus not distracted during the execution of the assessments, which might eventually affect the execution time. This also provides the benefit of integrating the display into the main system package and making it more appropriate for settings, with an open measurement space.

### 2.3. Conclusive Usability Study

As described in Section 3.1, USS usability was evaluated over 6 months in a conclusive study alongside the longitudinal TUMAL study [30] with a subgroup of the TUMAL cohort. The TUMAL study included 92 participants aged between 73 and 89 (average: 77.87, SD: 3.57) years, 51% of which were female, in a subgroup of the AEQUIPA Versa study [41].

Evaluation occurred at two time points during the TUMAL study (under the protocol in Figure 8): (1) Firstly, 15–30 min guideline-based telephone interviews were conducted between the second and third measurements on initial experiences (including barriers/difficulties/problems) with the USS. (2) Final guided focus groups were held shortly before or after the completion of the TUMAL study. (3) After the focus groups, participants were asked to complete a final questionnaire. The criteria for the group assignments included gender and the time of the last measurement.

One focus group contained only female subjects, two only male, and three represented both genders. The focus group lasted a maximum of two and a half hours, including a break. Different methods were used throughout, as explained in the Results section. To increase the quality of the qualitative data, important aspects were written on flipcharts during processing and summarized for all participants and, if necessary, missing or misunderstood information was added and corrected.

For qualitative research, the focus was on comprehensibility under Mayring [51] (e.g., a structured flow model during analysis and flow chart with questions during the focus groups) and on reliability under Kuckartz (e.g., application of a category system) [52]. At the end of the focus groups, participants were asked to complete a final questionnaire covering descriptive parameters such as gender, age, school, education, physical activity, technology use, technology experience (own scales), technology readiness [53], SUS [49], and the user experience questionnaire (UEQ short [54]).

The focus group discussions were evaluated content-analytically under Mayring [51]. Category formation was both inductive and deductive computer-aided via MAXQDA version 11 for Windows. After the text segments were assigned, the category system was discussed and validated with sample citations by the research team. Quantitative data from the questionnaires were calculated and analyzed descriptively using SPSS version 23 for Windows. The written notes from the telephone interviews and focus group discussions will be summarized together in a Word document in pseudonymous form. The questionnaire was designed objectively (e.g., explanation of the instrument, closed-ended questions, coding schedule) and validly (development process through literature review, revisions by project partners, semi-open-ended responses).

The TUMAL study is registered at the German Register for Clinical Trials (ID DRKS00015525) and approved by the medical ethics committee of the University of Oldenburg (ethical vote: CvO University Oldenburg medical ethics committee No. 2018-046) per the Declaration of Helsinki. The usability study is approved by the medical ethics committee of the University of Oldenburg (ethical vote: CvO University Oldenburg medical ethics committee No. 2018-094) per the Declaration of Helsinki.

## 3. Results

### 3.1. Cohort of the Conclusive Usability Study

Of the TUMAL cohort, 38 participants (34%) attended the usability study voluntarily. A total of 36 participated in a telephone interview (after 2 months), 32 participated in a focus group (shortly before the last measurement or shortly after divided into six groups), and 37 completed a questionnaire. The participants’ age ranged from 74–89 years (mean 79.41), 16 (43.2%) were female, and none had a migration background. Most participants in the study were married (n = 22). For highest school-leaving qualifications, 11 held a secondary school certificate. Multiple responses were possible when naming the highest level of education and university degree. Half had completed vocational—in-company training (n = 19). The other degrees were widely distributed (see Table 1).

In terms of technical device use, 28 participants reported using a PC/laptop or tablet either several times (n = 20) or once a day (n = 8). Fifteen individuals did not own a PC/laptop or tablet (see Table 2). Over half used a smartphone either several times (n = 19) or once a day (n = 2). Instead of, or in addition to, according to their statements, they owned a classic cell phone used several times (n = 12) or once a day (n = 1). Six and eight participants, respectively, stated that they did not own a classic cell phone or smartphone.

In response to the question “How confident do you feel about technology in general?” participants rated themselves as somewhat confident on a five-point Likert scale (1 = very unsure to 5 = very confident) with a mean of 3.54, covering the entire range, with an SD of 1.067. The cohort thus represents the full scale of experience with technologies, and participants rated their general technology readiness on a 5-item Likert scale with a mean of 3.09 and an SD of 0.397, where answers ranged narrowly between 3 and 4.

### 3.2. Usability Study

The median SUS score of 90 suggests good to very good usability. The interquartile range in the final evaluation is between 78.6 and 97.5, with a lower whisker score of 50 and an upper of 100 (see Figure 5). Five of the six UEQ subscales were rated between average and excellent. Participants rated the perspicuity subscale highest (MW: 1.953; SD: 1.046; CI: 0.337; 1.616–2.290) and the originality subscale lowest (MW: 0.270; SD: 1.126; CI: 0.363; −0.0930.632) (see Figure 9 and Table 3).

Asking focus group participants to summarize their USS experience by placing a dot on a 5-item Likert scale reported a mean value of 4.3 (with 5 representing “good”). The general positive attitude towards the USS was thus confirmed, with the response regarding the question of whether participants would intend continuous use of the USS providing 33 out of 36 approvals.

### 3.3. Results of the Telephone Interviews

A total of 36 participants stated that the instructions on using the USS were understandable, with 5 mentioning that it was easy to perform. Individual participants emphasized that they did not need the instructional videos because they found the operation to be very intuitive. A total of 29 suggested that they needed no assistance on site. Individual participants reported assistance with registration and logging in (especially where previous participants had not logged out) (4), challenges with connecting the sensor belt (3), software responsiveness (2), and assistance with the first measurement (1). Barriers cited below were the height of the chair (2) and sensor belt (1), which were described as being too short.

### 3.4. Insights of Focus Group Discussion

Considering the discussions of the six focus groups, insights were most relevant and are thus discussed in the following subsections.

#### 3.4.1. Experiences: RFID Chip/Reader, Cabin Construction, Chair, and Sensor Belt

The topic was worked on helped by a flipchart, on which positive experiences and suggestions for improvement were written and visualized by the moderator before being divided into the categories “RFID chip/reader”, “cabin construction”, “chair”, and “sensor belt”. Figure 10 shows the category tree with the number of assigned coded text passages.

RFID chip/reader: It was emphasized that everything worked, and the registration was clear (16 coded text segments). Most negatively mentioned aspects related to the registration not working at the beginning of the study, indicating that help was needed from the team. It was also unclear to some whether they were correctly registered (as the RFID reader confirmed successful login via a red LED) or whether they had to log out again after the assessments (22 coded text segments). As suggestions for improvement, these aspects could be derived: a small display with an indication that the registration succeeded, greeting as voice control, and a reminder to log off (19 coded text segments). With the initially requested active logout through a second activation of the RFID scanner, two participants suggested an automatic logout, which we then implemented.

Construction of the cabin: Overall, the participants considered the design of the prototype to be appropriate and sufficient for the study (15 coded text segments). This was contrasted by opinions that the cabin design was too simple and visually unattractive. Some participants found the passageway in the booth too narrow or described the curtain as unsanitary (16 coded text segments). In addition to color adjustments and a booth construction system, participants wanted, for example, an emergency call if something happened (28 coded text segments).

Chair: The chair was rated as functional and relatively comfortable (16 coded text segments). Other participants found the chair too high and its seat too low. Some said that it reminded them of an electric chair (nine coded text segments). To improve, the participants mentioned that it should be adjustable in height with a slanted seat to make it easier to stand up. The backrest should also be positioned further forward (five coded text segments).

Sensor belt: Positive comments were made about the sensor belt, including how it fit without problems and that the Velcro fastener worked with ease (eight coded text segments). Conversely, participants mentioned difficulties for older people, such as putting the belt on correctly or connecting the cable, which many felt was too small. Other participants mentioned problems putting it on if, for example, they had a higher BMI (29 coded text segments). As improvements, participants emphasized that the connector should be the other way around and larger, with a longer cable, and wireless charging. Other statements were related to handling the belt application (easier, instructions for use) or that the belt should be wider and longer overall (24 coded text segments).

#### 3.4.2. Experience Menu

The topic was dealt with using a flip-chart, on which positive and negative experiences and suggestions for improvement were written and visualized by the moderator before being divided into the categories of “operation”, “clarity”, “formulation/understandability of instructions”, and “design/optics”. Figure 11 shows the category tree with the number of assigned coded text passages.

Operation: The operation of the display was predominantly described as simple and understandable (16 coded text segments). Two negative experiences referred to the screen working hesitantly or not at all (two coded text segments). As improvements, the participants mentioned that the screen should not be on the side but straight ahead and operated via a remote control (three coded text segments).

Clarity: The clarity of the menu was described as clear (eight coded text segments). One participant mentioned there might be an inhibition threshold for users who are unfamiliar with the system (one coded text segment). Two improvements were provided on the color of the start button in the menu, which should be more clearly marked.

Formulation/understandability of the instructions: The formulation and understandability of the technology-supported instructions were predominantly reported as loud and clear (18 coded text segments). The male announcement voice was perceived as rather negative, and the start signal as somewhat too quiet (eight coded text segments). The following improvements were suggested: adjustability of the gender of the voice (individually adjustable) and volume (15 coded text segments).

Design/optics: The design and appearance were predominantly described as positive. According to participants, the font was easy to read, large enough, and the structure of the menu was recognizable (16 coded text segments). The negative aspects referred to the small step size and pictures or graphic sequence of the assessments (five coded text segments). The three suggestions for improvement related to the graphical representation of the measurements, which should be the other way around, and for one participant, the font and pictures should be somewhat larger (three coded text segments).

#### 3.4.3. Further Ideas for Enhancements

Various ideas for further development were derived from the six focus groups in a joint brainstorming session. The participants then rated the ideas with points, which were then discussed together. The number of points was based on the number of answers derived (87 coded text segments). Frequently mentioned questions and functions included feedback on user health status performance: “How participants are doing, whether they are healthy/performing, and what the wellbeing is like” (20 mentions), followed by a questionnaire to cover sport participation and, if so, what kind and how frequent (15 mentions). Technical enhancements included an extension of the prototype with hand strength measurements, a wobble plate (five mentions each), measurement of weight and height (four mentions), and measurement of breathing (one mention each). Further information can be found in Table 4.

#### 3.4.4. Appropriate Forms of Feedback of Measurement Results

Participants expressed the desire to receive digital feedback on their performance during the USS assessments. Visualization of their results and statistics was recognized as suitable (nine mentions), as was a comparison of personal and mean user performances in relevant age groups (two mentions). To identify appropriate forms of feedback on the measurement results, ideas were developed with participants based on a joint brainstorming session, followed by a subsequent evaluation of the data using the multipoint query and discussion approach (82 coded text segments). The participants favored feedback of the results via a homepage/portal (17 mentions), e-mail (16 mentions), as a printout onsite with comparison (12 mentions), a phone app (1 mention), or after each measurement via a graphical display (9 mentions). Further information can be found in Table 5.

#### 3.4.5. Overall Impression

The last question asked participants to summarize their overall impression. This was largely positive, and the measurements were perceived as easy, simple, and fast. The good support offered by the study team and organization was emphasized (14 coded text segments). Negative comments included the low volume of measurements in relation to effort, how participants were underchallenged and would have liked more measurements, and that the measurement box could be improved (nine coded text segments).

## 4. Discussion

Due to the participatory design, users were engaged intensively at various points before, during, and at the end of the TUMAL study, with various qualitative and quantitative methods considered a strength and recommended in technology evaluations [55,56]. Personal referral for participation by the study team and an advertisement poster at the USS motivated 38 participants from the TUMAL cohort to participate in the final evaluation. It can primarily be assumed that persons who noticed negative aspects and provided suggestions for improvement wanted to pass on or had an interest in receiving interim results. Brainstorming on enhancements led to intensive discussions in all focus groups and created many ideas for further USS improvements, underlining the participatory approach of this study. This target group should also be intensively involved in the further development of this technology.

Overall, participants reported being able to use the USS in their monthly assessments mostly without requiring assistance. We see this as a tremendous success as it assures the applicability of unsupervised assessment systems for older adults.

Compared to the first prototype, usability was rated significantly higher based on the SUS (with an enhancement from 75 to 90). This significant enhancement confirms the benefit of the chosen iterative participatory design approach. The individual values indicate the different positive and negative experiences. Lower variability in the individual SUS scores for the conclusive prototype indicates an overall better experience. However, when interpreting the results, it should be noted that the original usability study was conducted with 10 people and the final evaluation with 38. The duration of the studies also varied. The first SUS survey took place after a single test, and the second after the fourth or fifth measurement.

Ease of use was assessed using the UEQ in five subscales ranging from “average” to “excellent”. The low score in the “originality” subscale corresponds to feedback from participants who indicated that the assessments were too few and simple and that they would like to see more features.

With the RFID chip/reader, many negative aspects and suggestions for improvement were given, especially at the beginning of the study, where the login did not work, or the chip had to be held several times, which led to the need for technical support (e.g., restarting the computer). With the aTUG chair, besides reporting largely positive experiences, some participants suggested that a height-adjustable design was essential to ensure they could reach the floor and deliver a good test performance. It is also noticeable that some associated the chair with negative images (electric and medical chair), which should be remedied by design alterations.

With the sensor belt, difficulties in plugging in the USB port were mentioned, which could indicate age-related limitations such as reduced vision or impaired motor and sensory functions. Sensory and motor functions play a central role in using and integrating technical devices. Sensory functions in particular (e.g., vision, touch) may decline with age [57]. After 70, most people are affected by changes in their sense of touch [58]. A wireless charging and data transfer mechanism proposed by the participants could overcome these barriers and is considered an appropriate approach here. Although the strap was already lengthened in the usability study, some participants, especially those with higher body weight, still felt it was too narrow and should be lengthened again.

While many participants found the operation of the display and menu to be positive, some reported challenges, which likely resulted from their relative experience with technology. One person stated they owned a computer/laptop, a tablet PC, or a smartphone, but never used these devices in everyday life. In other studies, the user-friendliness of systems and technical experience are also seen as important factors promoting successful use [59].

With the wording and comprehensibility of the instructions, individual peculiarities were the main topic of discussion. While some perceived the sounds as too quiet, others were critical of the signal tone, regarding it as too loud or long. Due to the longer duration of the study, recall bias cannot be ruled out. Although the group interviews were conducted shortly before or after the end of the study, some participants could not recall certain questions or functions.

Although many participants reported negative experiences and suggestions for improvement, this is certainly due to the methods used in the focus groups, which asked more critical questions about the components of the prototype. Some participants were also disappointed during the group sessions not to receive individual results and recommendations from the measurements.

Overall, the detailed explanation and briefing by the study team had a positive impact on the use and application of the USS. Likewise, the quick accessibility maintained effectiveness in case of problems and difficulties. This influence should be further investigated in continuing studies at other USS sites, especially when technical support is not immediately available in person and participants receive no briefing beforehand. Here, additional consideration should be given to the need to adapt the design of the USS to the environment. In a recent fitness studio installation, for example, we reduced the design (including removing the walkway’s wood paneling) to meet spacing requirements. The resulting version is shown in Figure 12.

For the conclusive prototype, visualization and feedback of subjects’ assessment results were excluded under the Medical Devices Regulation (MDR) and corresponding requirements of the ethics board were applied. User experiences when interpreting assessment results thus remain an open research topic for later enhancements.

Study design weaknesses can be observed in the unintentionally selective composition of the target group. Overall, participants were generally active in everyday life (e.g., sports club) and engaged in various topics (participation in other studies). No participant had a migration background, and with technology experience and use, it was found that the majority regularly used technical devices and on average considered themselves confident with them.

Other studies in the field of technology development also report on selective target groups and the challenges that come with including others in the development process [60,61]. In future studies, it is important for other groups (e.g., inactive people, the very old, people who are not technology-savvy, people with a migration background) to be involved, reflecting the heterogeneity of age. The use of technological devices depends on several factors besides age, such as gender, socioeconomic status, and cultural background [62,63,64]. To this end, participants discussed ideas and suggestions about which groups of people might benefit from these measurements and where future prototypes should be located to promote their use. Inactive people were frequently mentioned here. Future locations for a measurement box were given as care facilities, doctors’ surgeries, department stores, or cultural centers.

## 5. Conclusions

This article presents the user interfaces of the unsupervised screening system (USS), which performs an unsupervised assessment of TUG and 5CRT in older adults for early detection of functional decline. After confirming the sensitivity to measure TUG and 5CRT performance in the TUMAL study [30], this article focused on the usability and user acceptance of the system. As part of a participatory design process, user acceptance and usability of an early prototype of the USS were investigated, and the user interface improved. Overall, usability was significantly improved compared to the first prototype. Although many positive experiences were reported, improvements for further development of the USS could also be derived. These include individually adjustable components such as the chair, sensor belt, and functions such as font and image size, volume, signal tone, and announcement voice. Above all, age-specific characteristics and limitations must be considered and show how important the active participation of this target group is in the development process. A central issue for participants was the lack of feedback on test performance, which was excluded in this study due to restrictions imposed by medical device regulations. The evaluated prototype offers a high potential for early detection of functional impairment in the elderly and could be extended by additional functions. The measured values could be used by therapeutic and health professionals for preventive measures to counteract deterioration and loss of mobility. The independent use of the measurement system could reduce the personnel and organizational effort in contrast to conventional procedures and thus relieve the healthcare system.

To address older groups, future studies should consider other access routes for recruiting hard-to-reach target groups. These include multipliers from the community, target group, and church institutions. This way, elderly individuals with a migration background and people who are immobile, very old, or not tech-savvy could also be included, and subsequently benefit from this technology. In the future, the USS should be located where the everyday life of these people occurs. In the area of prevention and health promotion, this includes close social environments, such as a local neighborhood or district. Future studies are intended to investigate these aspects.

## Figures and Tables

**Figure 1 sensors-22-00731-f001:**
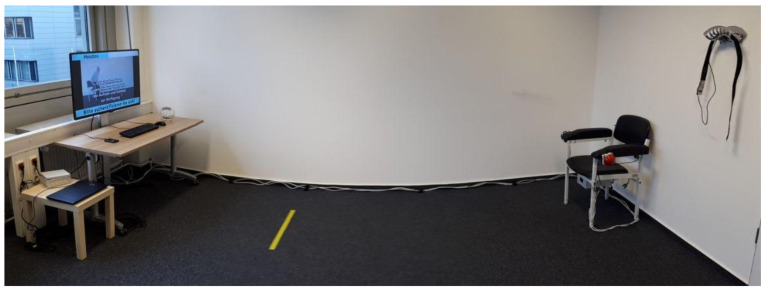
The Initial prototype comprises the main components of user interaction within the system and has been used for the user interaction study.

**Figure 2 sensors-22-00731-f002:**
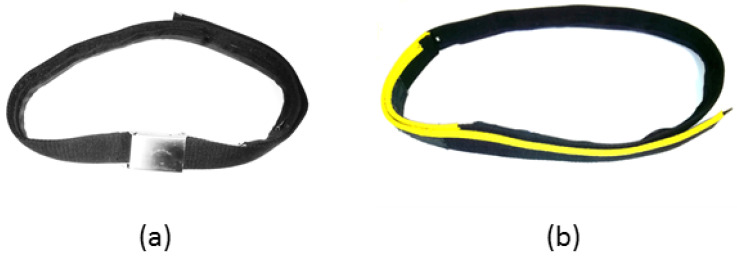
The applied sensor belt (**a**) in its original design and (**b**) in its adapted form.

**Figure 3 sensors-22-00731-f003:**
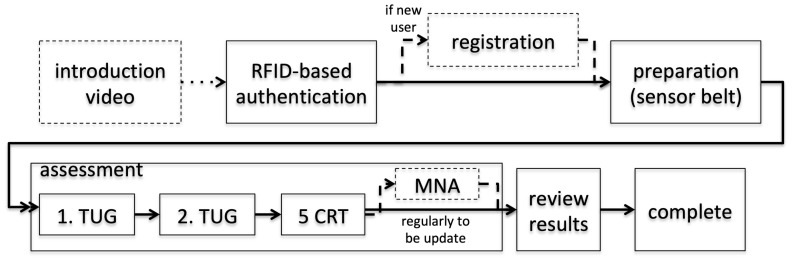
System’s user-interaction workflow; incorporating user-interaction phases and assessments.

**Figure 4 sensors-22-00731-f004:**
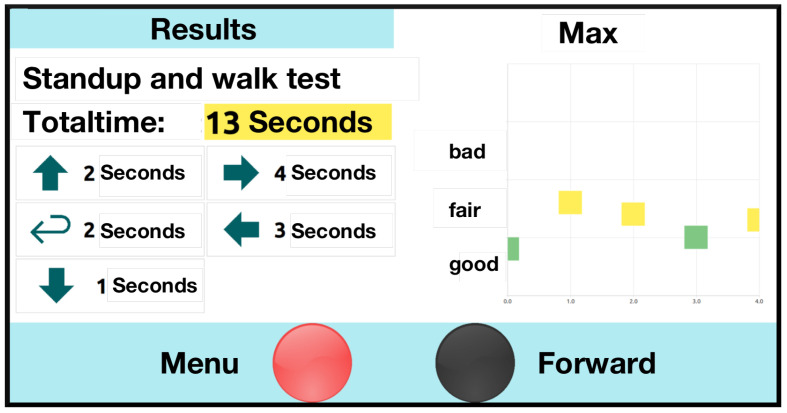
The initial results presentation for the TUG test combines the overall test duration, the duration of the relevant iTUG sub-phases, and a color-encoded presentation in a timeline (on the right).

**Figure 5 sensors-22-00731-f005:**
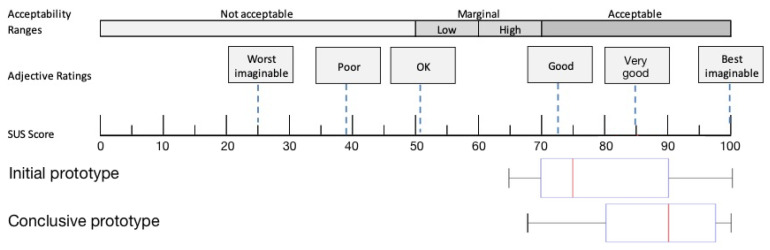
System usability score of the initial and conclusive prototype shown as horizontal box plots (initial prototype n = 10, conclusive prototype n = 33).

**Figure 6 sensors-22-00731-f006:**
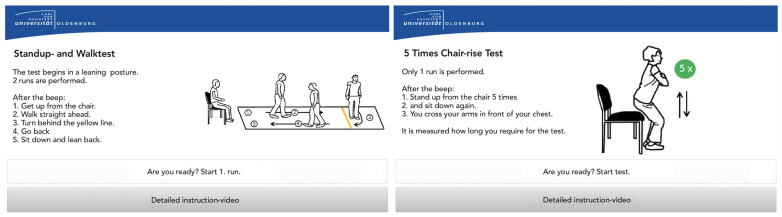
Examples for the instructional screens for the TUG and the 5 CRT. As the prototype has been only evaluated and implemented with German native speakers, the screens have been translated into English for this article.

**Figure 7 sensors-22-00731-f007:**
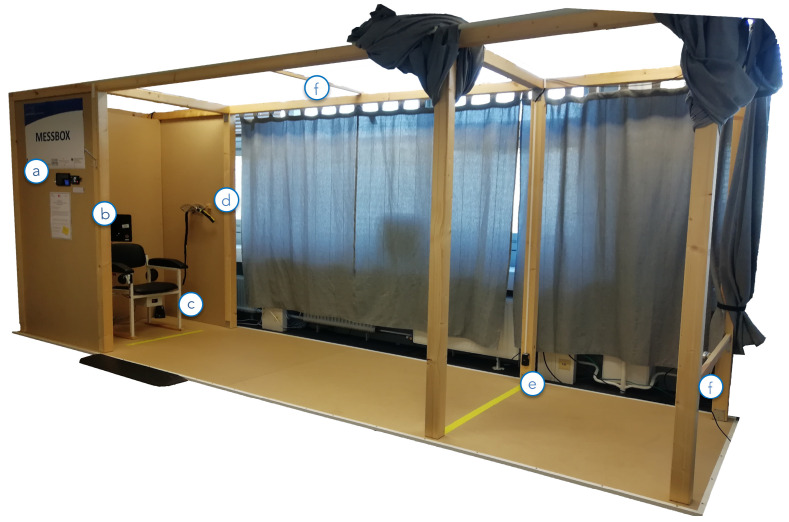
The evaluation prototype comprising (a) an introductory display and an RFID authentication device, (b) main display for user interaction, (c) the integrated aTUG chair with additional LBs, (d) a sensor belt including an inertial sensor, (e) light barriers at 3 m walking distance, (f) Intel RealSense D435 depth image camera placed approximately 4 m from the chair at an upper level facing onto it. Image originally published in [30].

**Figure 8 sensors-22-00731-f008:**
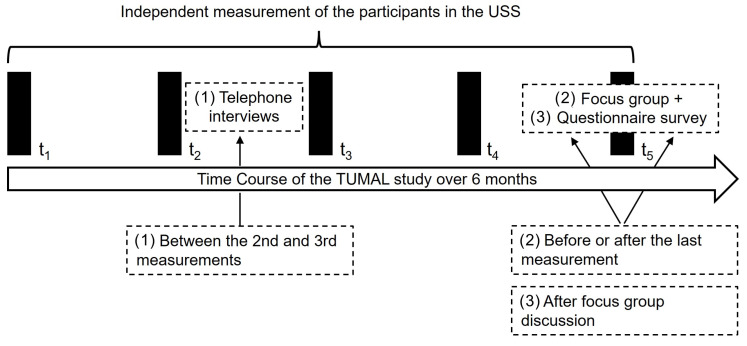
The protocol of the TUMAL study.

**Figure 9 sensors-22-00731-f009:**
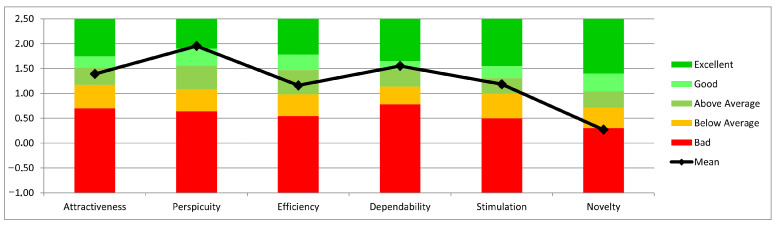
User experience questionnaire (n = 37).

**Figure 10 sensors-22-00731-f010:**
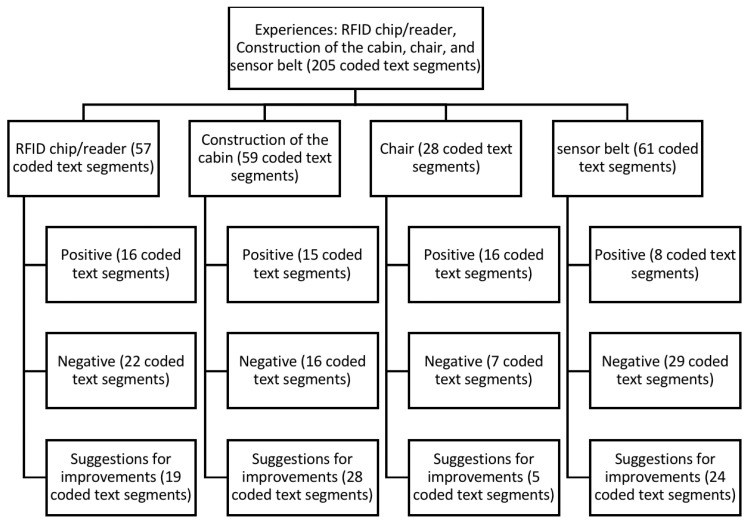
Summary representation of the focus group interviews based on the coded text segments related to RFID chip/reader experiences and construction of the cabin, chair, and sensor belt.

**Figure 11 sensors-22-00731-f011:**
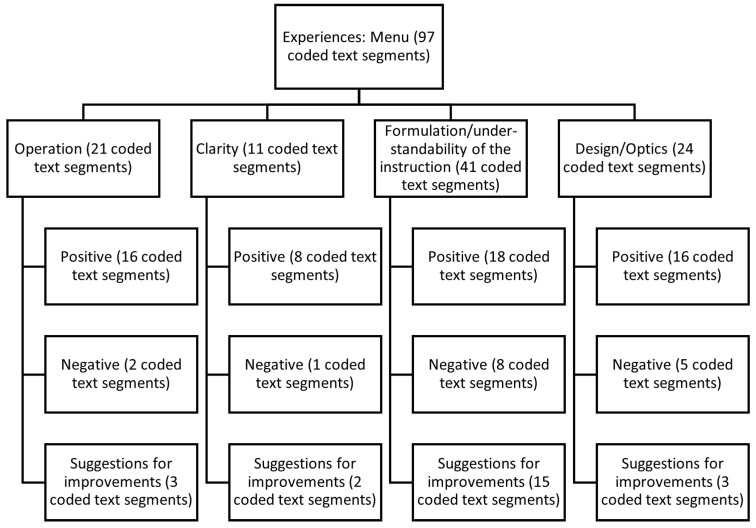
Summary representation of the focus group interviews based on the coded text segments related to the user menu.

**Figure 12 sensors-22-00731-f012:**
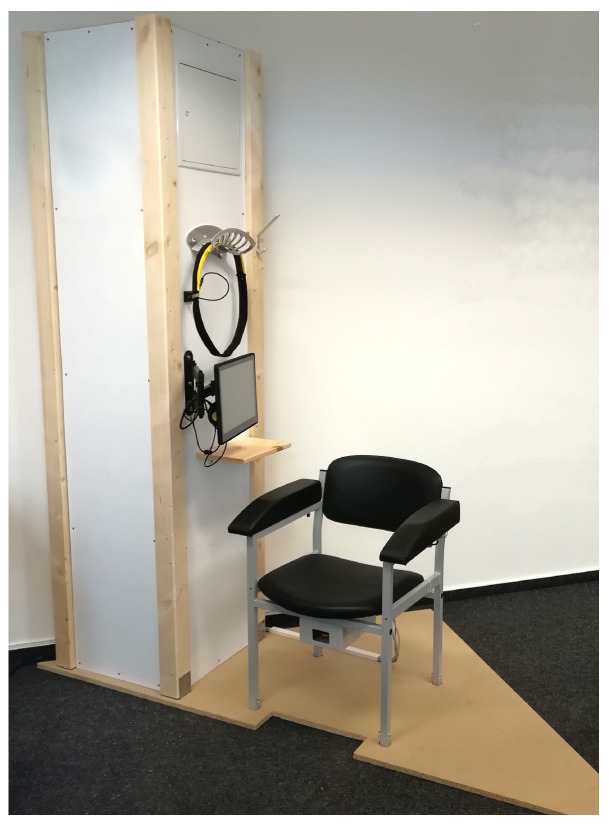
Example of a later USS, where the casing was adjusted for the requirements of a sports club.

**Table 1 sensors-22-00731-t001:** Additional sociodemographic parameters (frequency as n), ordered in accordance to occurrence; regarding training/university degrees, multiselection was supported (n = 37).

Parameter	Item	Occurence
Marital status	Married, living with spouse	22
	Widowed	10
	Divorced	3
Highest school degree	Secondary school certificate	11
	Secondary/elementary school	6
	University entrance qualification/secondary school	6
	Advanced technical college entrance qualification	1
Training/university degrees	Vocational—in-company training	19
	Vocational—school education	8
	Technical college/engineering school	5
	University/college	5
	Technical school, master school, technical school,	3
	vocational or technical academy	
	Other educational qualification	3
	None	1

**Table 2 sensors-22-00731-t002:** Frequency of technical device usages, describing the cohorts technological experience (n = 37).

	How Often Do You Use the Following Technical Devices?					
	**PC/**	**Tablet**	**Classic**	**Smart**	**Smart**	**Other**
	**Laptop**	**PC**	**Bar Phone**	**Phone**	**Watch**	**Device**
Several times a day	16 (43%)	4 (11%)	12 (32%)	19 (51%)	0 (0%)	0 (0%)
Once a day	8 (22%)	0 (0%)	1 (3%)	2 (5%)	0 (0%)	0 (0%)
Several times a week	4 (11%)	2 (5%)	2 (5%)	1 (3%)	0 (0%)	0 (0%)
Once a week	3 (8%)	1 (3%)	0 (0%)	0 (0%)	0 (0%)	0 (0%)
Less than once a week	2 (5%)	5 (14%)	4 (11%)	0 (0%)	0 (0%)	0 (0%)
I own but never use	1 (3%)	1 (3%)	2 (5%)	1 (3%)	0 (0%)	1 (3%)
I do not use	2 (5%)	13 (35%)	6 (16%)	8 (22%)	21 (57%)	14 (38%)
Total	36 (97%)	26 (70%)	27 (73%)	31 (84%)	21 (57%)	15 (41%)
Missing	1 (3%)	11 (30%)	10 (27%)	6 (16%)	16 (43%)	22 (60%)

**Table 3 sensors-22-00731-t003:** Scales UEQ confidence intervals (*p* = 0.05) per scale (n = 37).

Scale	Mean	Std. Dev.	N	Confidence	Confidence Interval	
Attractiveness	1.390	1.189	37	0.383	1.007	1.774
Perspicuity	1.953	1.046	37	0.337	1.616	2.290
Efficiency	1.162	1.061	37	0.342	0.820	1.504
Dependability	1.550	0.974	37	0.314	1.236	1.864
Stimulation	1.186	1.184	37	0.382	0.804	1.567
Novelty	0.270	1.126	37	0.363	−0.093	0.632

**Table 4 sensors-22-00731-t004:** Derived questions and functions for further development.

Questions/Functions	Total Mentions
How are you doing healthwise? Has anything changed?	20
Do you feel healthy and able?/Wellbeing question	
Do you participate in any sports (if yes, how often)?	15
How did you obtain the measuring box?	6
Do you currently use assistive mobility devices?	5
Hand force measurement	5
Wobble plate for balance measurement	5
Comments/improvement suggestions after measurement	4
What would help you maintain your performance?	4
Automatic measurement of weight and height	4
Question about taking medication	4
Fatigue question	3
Has anything changed since the first question until today?	3
Relaxation exercise	2
Display data after use	1
Have you been ill between measurements?	1
(If yes, do you wish to talk?)	
How do you feel involved in your environment?	1
Balance measurement camera	1
Are you a smoker?	1
Do you drink alcohol?	1
Breathing measurement	1

**Table 5 sensors-22-00731-t005:** Ideas for enhancements.

Ideas	Total Mentions
Homepage/portal	17
E-Mail	16
Printout on site (with comparison)	12
After each measurement review of the results of the measurement	9
(graphical display, statistics)	
Personal conversation	7
Closing/information event	6
Group meeting with exchange on a specific topic	3
Feedback with reference average value of the respective age group	2
Personal contact after last measurement	2
Feedback directly from the doctor	2
Feedback via app	1
Written report	1

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
