# Peer review of "Measurement System for Unsupervised Standardized Assessments of Timed Up and Go Test and 5 Times Chair Rise Test in Community Settings—A Usability Study"

_sensors, 2022, doi:10.3390/s22030731_

Round 1
Reviewer 1 Report
The manuscript presents an investigation into the usability of an unsupervised assessment system (USS). The system itself has been reported in a previous publication. An initial prototype was evaluated in experiments comprised of three stages in which 10 older adults participated. Stage one was an introduction to the system. Stage two was a task-based evaluation in which participants tried to solve three distinct tasks related to the use of the system. The third and final stage was a semi-structured interview to extract the participant's experience. Based on results from the evaluation of the initial prototype a conclusive prototype was developed. This prototype was evaluated in a combination of phone interview, questionnair and focus groups. A group of 38 older adults participated.   
* Major comments
- The questionnairs and other materials used for the evaluations should be made available as supplementary material.
- The main weakness of the study is the fact that the sample is not representative of the complete target user population. The authors discuss this adequately, though, and the findings will still have value. 
* Minor comments
- Several paragraphs consisting of a single sentence. Check if sentence can be included in previous or following paragraph.
- line 39: 5TCR should be 5CRT
- line 66: Unclear what is meant by "personal approach" in this context. Is it "individualized care"?
- line 164: Use "was evaluated" to be consistent.
- lines 183-185. How was the concept of "suitability" explained to the participants?
- line 195: Mean and standard deviation are not appropriate with 10 participants in my opinion. Median and range should be sufficient.
- line 210: Write "identifying" instead of "locating"?
- lines 216-218: "Attempts to enter data in phases ..." This is unclear. Could an example be provided?
- line 237: Better to directly write "increased waist (or chest) width" instead of the indirect "increased body-mass index".
- line 301: What does "ow" stand for?
- line 354: "permeability" should be "persipicuity".
- Figure 9: "Persipicuity" is not a particular common word in English (but maybe more so in German?). How was the concept explained to the participants?
- line 414: What is an ip-chart?
- line 443-444: What is meant exactly by "multi-point query"?
- line 505: "oor"?
Reviewer 2 Report
Dear Ms. Pauline Yang,
Thank you very much for the possibility to serve as a Reviewer in a prestigious periodical like Sensors.
About this paper, the contents are appreciable and the paper is well organized.
Summary of the research and overall impression
In the present manuscript, the authors evaluate and enhance the usability of the Unsupervised Screening System (USS), introduced in community-dwelling older adults, for the assessment of dynamic balance and lower limb muscular strength through Time Up and Go and 5 Time Chair Rise Test.
The authors well described all the usability-study phase. In my opinion, the paper can be published in this form.
Reviewer 3 Report
Dear authors first of all congratulate you for investing in a very challenging area and with the increasing percentage of elderly people in the developed world it becomes more and more important.
In my opinion there are only two or three important questions, the first a brief explanation of TUG and 5 Times Chair Rise Test.
Phases and how are important. I know it's probably already in the previous article but it's important.
Afterwards, I also understand that it would be important to analyze other approaches by other authors and make a comparison with this one.
Finally, perhaps more explicitly to summarize the next steps of the study.
